# Effectiveness of the SA 14-14-2 Live-Attenuated Japanese Encephalitis Vaccine in Myanmar

**DOI:** 10.3390/vaccines9060568

**Published:** 2021-05-31

**Authors:** Mya Myat Ngwe Tun, Aung Kyaw Kyaw, Khine Mya Nwe, Shingo Inoue, Kyaw Zin Thant, Kouichi Morita

**Affiliations:** 1Department of Virology, Institute of Tropical Medicine and Leading Program, Nagasaki University, Nagasaki 852-8523, Japan; bb55418851@ms.nagasaki-u.ac.jp (K.M.N.); pampanga@nagasaki-u.ac.jp (S.I.); moritak@nagasaki-u.ac.jp (K.M.); 2Department of Medical Research, Pyin Oo Lwin Branch, Pyin Oo Lwin 05082, Myanmar; aungkyawkyaw@mohs.gov.mm; 3Myanmar Academy of Medical Sciences, Yangon 11181, Myanmar; kyawzinthant@mohs.gov.mm

**Keywords:** SA-14-14-2 vaccine, Japanese encephalitis virus, children, catch-up campaign, Myanmar

## Abstract

Myanmar is an endemic country for the Japanese encephalitis virus (JEV), and the SA-14-14-2 live-attenuated JEV vaccine was first introduced as a catch-up vaccination campaign in 2017. To determine the effectiveness of vaccination by means of neutralizing antibody titers against JEV, a cross-sectional descriptive study was conducted among five to 15-year-old monastic school children in Mandalay, Myanmar. A total of 198 students who had received vaccines were recruited, and single-time investigation of anti-JEV IgG and neutralizing antibodies against wild-type JEV were determined using anti-JEV IgG ELISA and plaque reduction neutralization tests (PRNT_50_). All students 100% (198/198) showed positive results on the anti-JEV IgG ELISA, and 87% (172/198) of the students had neutralizing antibodies against JEV six months after immunization. The geometric mean titers of both IgG antibodies and neutralizing antibodies increased with the participants’ age groups, and statistically significant differences in anti-JEV IgG titers were noted across age groups. In this study, we could not investigate the persistence of neutralizing antibodies as only single-time blood collection was done. This study, which is the first report of JEV vaccination among children in Myanmar, showed similar neutralizing antibody production rates among vaccinated individuals as did studies in other countries.

## 1. Introduction

Japanese encephalitis (JE) infection, a vaccine-preventable disease caused by a mosquito borne flavivirus, the Japanese encephalitis virus (JEV), is one of the leading causes of viral encephalitis in Asia. Approximately three billion people, including 700 million children, live in areas at risk for JE infection [1]. An estimated 70,000 cases are reported annually, and the World Health Organization (WHO) estimates that JE claims 14,000 to 20,000 lives a year, mostly children under 15 years of age. Among survivors, half to three-quarters exhibit long-term intellectual, behavioral or neurological disabilities such as paralysis or the inability to speak. No cure or clinical treatment exists for JE [2]. Because mosquito vector control is not yet sustainable or cost-effective, vaccination is the most important measure to prevent JE. 

Myanmar is among Asia’s JE-endemic countries due to its ecological environment, which is characterized by widespread rice farming, wading birds and pig production. These conditions are conducive to JE transmission through infected mosquitoes. JE infection was first documented in 1977, and sporadic JE outbreaks have been reported in Myanmar since then [3,4,5]. However, widespread outbreaks began in 2014 and continue to pose a significant public health problem. After gradual increases, the number of detected cases reached 404 in 2016 [6]. At present, JE cases have been detected in all states and regions of Myanmar, primarily affecting children aged nine months to 15 years old [7]. In Myanmar, the national Expanded Program on Immunization (EPI), which is supported by WHO and UNICEF, began a catch-up campaign in 2018 and has now made JE vaccination routine. This study aimed to evaluate the effectiveness of the catch-up JE vaccination regimen in Myanmar by assessing the neutralizing antibody among monastic school children in Mandalay, Myanmar. 

## 2. Materials and Methods

### 2.1. Samples and Vaccine

To explore the effectiveness of the SA-14-14-2 vaccine after a catch-up JEV vaccination campaign in Mandalay, Myanmar, a cross-sectional descriptive study was conducted in three monastic schools in Mandalay in 2018. A total of 198 apparently healthy students (five to 15 years old) who had received a single dose of the SA 14-14-2 lived-attenuated JEV vaccine were recruited. All students were vaccinated at the same time during the first phase of JEV vaccine catch-up campaign conducted by Ministry of Health and Sports, Myanmar. We calculated the sample size using the formula, N = Z^2^P (1 − P)/d^2^. We assumed the proportion of presence of neutralizing antibody among vaccinated students would be 40%, and precision was taken as 0.07% and 95% confidence interval. Single-time blood collection was performed six months after vaccine administration, and in-house anti-JEV IgG ELISA and a plaque reduction neutralization test (PRNT) which was validated in a previous study, were performed on all samples at Virology Department, Institute of Tropical Medicine, Nagasaki University, Japan [8]. In this study, the effectiveness of the vaccination program was determined by measuring neutralizing antibodies against JEV. 

### 2.2. Determination of Anti-JEV Antibody by ELISA

For the anti-JEV IgG ELISA, serum samples were diluted at 1:1000 dilution with phosphate-buffered saline containing 0.05% Tween (PBS-T). All of the 96-well microplates except the blank were coated with 100 µL of purified JEV antigen, whole virion (strain: JaOrS982; 250 ng/100 µL/well) and incubated at 4 °C overnight. With the exception of the blank, the wells were blocked with the original concentration of Blockace (Yukijirushi, Sapporo, Japan) and kept at room temperature (RT) for one hour. The diluted test sera and control samples (100 µL/well) were added to each well and incubated at 37 °C for one hour. Then, 100 µL of 1:30,000 diluted HRPO-conjugated goat anti-human IgG (American Qualex, San Clemente, CA, USA) in PBS-T plus 10% Blockace was added to each well before the wells were placed at 37 °C for one hour. Initiation of the peroxidase reaction occurred after incubation at RT for 30 min in the dark. The reaction was then halted by the addition of 100 µL of 1 N sulfuric acid per well. The plates were washed three times with PBS-T after the completion of each step. The optical density (OD) was read at 492 nm, and the IgG titers of vaccine recipients’ sera were determined from a positive standard curve. IgG titers greater than or equal to 1:3000 titers were considered positive [8]. 

### 2.3. Detection of Neutralizing Antibody to JEV

A plaque reduction neutralization test (PRNT) was performed to confirm the neutralizing antibodies (NAb) against JEV. First, the serum samples were diluted from 1:10 to 1:5120 dilution with a cell culture medium. The JaOrS982 strain was used in this study. One hundred and fifty µL of the diluted serum samples were mixed with an equal volume of virus, which included 50 plaque-forming units. This mixture was incubated at 37 °C for one hour to induce a virus-antibody neutralization reaction. Next, 150 µL of the diluted serum and virus mixture was inoculated on a Vero cell monolayer in a 24-well plate. After incubation at 37 °C for one hour, the infected cells were overlaid with 500 µL of 1.25% methylcellulose 4000 in 2% FCS MEM. The plates were then incubated at 37 °C for 4 days. The plates were washed thrice with PBS to remove the methylcellulose, fixed with a 4% paraformaldehyde phosphate buffer solution (Wako, Osaka, Japan) for 30 min at RT, rinsed, and stained with crystal violet stain. The neutralizing antibody was calculated based on a 50% reduction of the plaque count. The maximum tested sera dilution capable of achieving the 50% reduction was considered the neutralizing antibody level [5,9]. 

### 2.4. Statistical Analysis

GraphPad Prism (GraphPad Software Inc., La Jolla, CA, USA) was employed for statistical analyses. Both IgG titers and neutralizing antibody (NAb) levels were expressed as mean and geometric means with a 95% confidence interval (95% CI). All comparisons among the four groups were performed by Anova test, and the Mann-Whitney U test was used to compare IgG Ab and NAb titers among different age groups. The presence of neutralizing antibodies at a 50% plaque reduction neutralization titer (PRNT_50_) of at least 1:10, which has been established as a correlate of protection against the development of JE in humans, was defined as the cutoff point in this study [10].

## 3. Results

### 3.1. Anti-JEV IgG Titers

All 198 students (100%) showed positive results on the anti-JEV IgG ELISA, but only 172 (87%) students had neutralizing antibodies against JEV, and 26 students (13%) had less than 1:10 neutralizing antibodies against JEV. Of the 198 students, 90 students were male, and 108 were female, representing a male-to-female ratio of 5:6. The mean age of the students was 12.29 years (standard deviation, SD: 2.7 years). 

We compared the mean and geometric mean antibody titers (GMT) of anti-JEV IgG and NAb. The mean and GMT for all students were 17,072 (95% CI, 15,061–19,083) and 12,335 (95% CI, 11,003–13,829), respectively. The minimum anti-JEV IgG titers equaled 3013, whereas the maximum titers reached 72,169. The GMT of the JEV IgG antibodies gradually increased with age groups. The GMT of the JEV IgG was the lowest (6046, 95% CI, 3826–9553) in the youngest age group (5–7 years) and the highest (15,468, 95% CI, 12,710–18,824) in the oldest (14–15 years) age group (Table 1). The mean antibody titers between the four age groups were statistically different (*p*-value = 0.0009) in the two-way ANOVA analysis. A Mann–Whitney U test was performed to determine differences between each age group, and these differences were found to be statistically significant (*p* values of <0.001, <0.01, <0.05; Figure 1).

### 3.2. Neutralizing Antibody Titers 

The mean and GMT of NAb for 172 students were 93 (95% CI, 33–154) and 28 (95% CI, 23–34), respectively. Similarly, the mean and GMT of the NAb also exhibited an increasing trend with age (Table 1). The lowest and highest means and GMT of neutralizing antibodies were found in the youngest (5–7 years) and oldest (14–15 years) age groups, respectively. Eighty-two students (41%) had NAb titers of 1:10, and only one student (0.5%) had neutralizing antibody titers of 1:5120 against JEV (Figure 2). One percent to 11% of students possessed NAb titers between 20 and 640.

## 4. Discussion

Vaccination is the primary strategy for prevention of infectious diseases and JEV vaccination has proven extremely effective in reducing the incidence infections, particularly in South Korea [11,12], Japan, China [13] Thailand [14] and Taiwan [15]. In Myanmar, the JEV vaccine (SA 14-14-2 live-attenuated vaccine manufactured in China) was firstly introduced in a catch-up vaccination campaign and added to the lists of the EPI programs receiving Global Alliance and Vaccine Immunization support at the end of 2017.

The vaccination campaign in Myanmar involved two phases. Phase 1 commenced at public and private schools targeting five-to-15-year-olds, whereas Phase 2 commenced in the community targeting children aged nine months to five years who were not attending school. During Myanmar’s first nationwide immunization campaign against JEV, vaccine recipients included more than 12 million children, covering 92.5% of the country. In Myanmar, the number of confirmed JEV cases also decreased after the JEV vaccine was introduced. From 383 in 2017, the total number of cases fell to 126 in 2018 and 115 in 2019 [16].

Utilizing serological and molecular methods, previous studies reported that genotypes I and III of the JEV were circulating in Myanmar [5,17]. The SA 14-14-2 vaccine strain, which uses genotype III of the JEV, also showed protective effects against both genotype I and III, but its protective effect against genotype V was lower. Although, therefore, useful for the prevention of disease, vaccination does not eliminate the need for molecular surveillance, which should be conducted annually to detect for the introduction of new genotypes (genotype V) to Myanmar.

To assess the effectiveness of the JEV vaccine, neutralizing antibodies measured by PRNT as a seroprotection rate were used as the end point for the vaccine study [18]. In this study, 87% of SA 14-14-2 JEV-vaccinated students had greater than 1:10 neutralizing antibodies against JEV, and 13% had less than 1:10. In a previous laboratory-based study, human serum samples from JEV-vaccinated people were tested on mice challenged with virulent JEV strains. These samples showed a titer greater than1:10 and had a protective effect; partial protective activity was also noted with titers less than 1:10 [19].

Many studies have reported the effectiveness of a single dose of SA 14-14-2 JEV vaccination at 85–95%, and booster doses can increase the efficacy to 100%. In this study, 87% of students had neutralizing antibodies. Moreover, some studies stated that the efficacy of the JEV vaccine differed across populations. In a study in South Korea and China, the vaccine achieved an efficacy of 85–95% [12,20], and one study conducted in Nepal found that a single dose of vaccine produced an efficacy of 98.5% one year after immunization [21]. In a study involving an Indian population, however, the efficacy was much lower according to a post-marketing serosurvey, which reported a seroprotection rate after 28 days of only 67.2% [22]. In this study, we evaluated the seroprotection rate of vaccinated children after only six months. Further research is, therefore, required to confirm the longevity of the JE vaccine’s protective response.

Although all cases (100%) in this study showed positive results on the IgG ELISA, some students had no neutralizing antibodies against JEV, perhaps due to the cross-reactivity of IgG antibodies with other flaviviruses. Indeed, IgG is extremely cross-reactive and can cause cross-reactivity [23]. Moreover, Myanmar is an endemic country for many flaviviruses, such as dengue and Zika. Besides, we used whole virion of JEV as the purified antigen and it could cause cross-reactivity to other flaviviruses.

Despite its contributions, this study has limitations. Our investigation involved only a one-time serum sample collected six months after vaccination and, therefore, could not explore the persistence of neutralizing antibodies. Further studies will be conducted to determine both the persistence of neutralizing antibodies and the immunologic response after booster doses, which are important factors for making political decisions for JEV vaccination via the EPI program.

## Figures and Tables

**Figure 1 vaccines-09-00568-f001:**
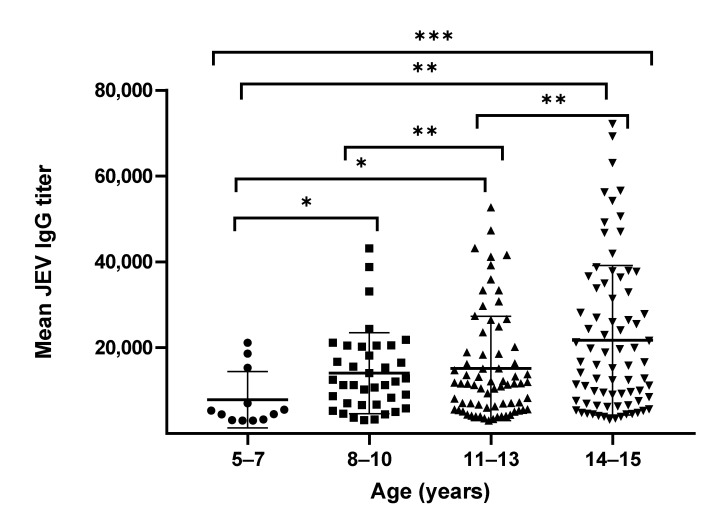
Distribution of anti-JEV IgG antibody titers among different age groups. A *p* value less than 0.05 is considered statistically significant (* *p* < 0.05; ** *p* < 0.01; *** *p* < 0.001).

**Figure 2 vaccines-09-00568-f002:**
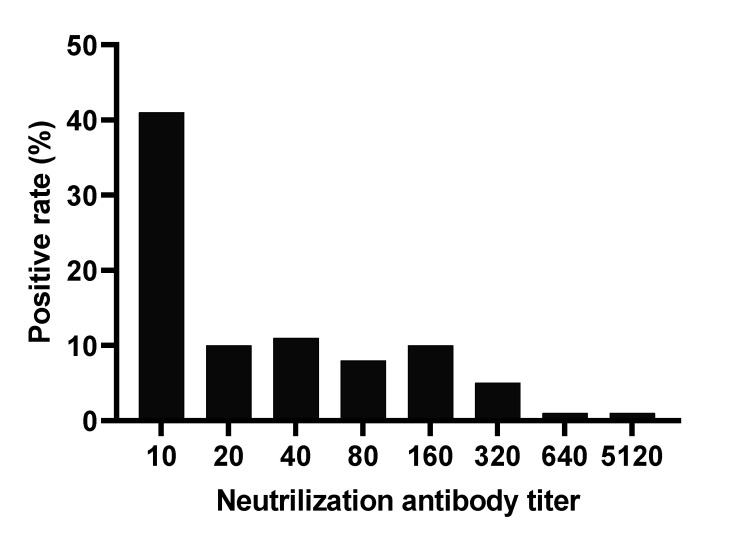
Proportion of anti-JEV neutralizing antibody titers among vaccinated students.

**Table 1 vaccines-09-00568-t001:** Geometric mean titers of Anti-JEV IgG and neutralizing antibodies against JEV.

Age Group (Years)	Gender (Female:Male)	Mean of JEV IgG Titer (95% CI)	GMT of JEV IgG Titer (95% CI) *	Mean of JEV NAb Titer (95% CI)	GMT of JEV NAb Titer (95% CI)
5–7	5:7	7876 (3705–12,047)	6046 (3826–9553)	22 (0–52)	14 (9–34)
8–10	18:20	14,082 (10,982–17,182)	11,401 (9133–14,233)	37 (0–74)	16 (12–22)
11–13	39:32	15,158 (12,282–18,034)	11,356 (9477–13,607)	153 (0–324)	30 (21–44)
14–15	28:49	21,746 (17,784–25,709)	15,468 (12,710–18,824)	76 (53–98)	37 (28–49)

* GMT of IgG Ab statistically different among different age groups (*p* values < 0.05). GMT = Geometric mean titers, NAb = Neutralizing Antibodies, 95%CI = 95% Confidence Interval.

## Data Availability

Not applicable.

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
