# Peer review of "Effectiveness of the SA 14-14-2 Live-Attenuated Japanese Encephalitis Vaccine in Myanmar"

_vaccines, 2021, doi:10.3390/vaccines9060568_

Round 1

Reviewer 1 Report

This study is a cross-sectional evaluation of JEV vaccine seroprotection via IgG ELISA and PRNT assays in children in Myanmar as part of a catch-up vaccination campaign. While this is a small study, this is an important topic and vaccine data and efficacy from the Greater Mekong Subregion, particularly Myanmar, Laos, and Cambodia, is lacking and should be published.

MAJOR COMMENT:

The authors point out the main limitation of the study, which is cross-sectional nature, and should be better emphasized in the abstract. The duration of nAb response will be critical to follow up.

Were all the children vaccinated at the same time or could you model the antibody decay by using different vaccination dates of the children? (e.g. 50 children were vaccinated 6 months prior to blood draw and some where vaccinated 2 months prior to blood draw, etc etc). There should be some type of sample size calculation so the reader can gauge the precision of your estimates given that 198 children is not a very large number, especially if this will then extrapolated nationally.

The 95% CIs are large in the table and are an indicator that there is not very much precision so it would be nice to see these calculations quantified somewhere.

Otherwise, I only have some minor comments for review:

  • Line 160-162: example - nomenclature for nAB - "students had greater than 10 nABs against JEV" - it should be titers of 1:10 - it is incorrectly written in a few places like this
  • - Line 178: for those positive on ELISA but negative on nABs, what is the likelihood of non-neutralizing antibodies being responsible? It talks about IgG being cross-reactive but more specificity is needed - is it presumably NS1 IgG that is cross reactive with other flavivirus NS1? It says purified JEV antigen in the Methods - is it ELISA against whole virus?
  • - Line 180: there is an extra "."

Author Response

This study is a cross-sectional evaluation of JEV vaccine seroprotection via IgG ELISA and PRNT assays in children in Myanmar as part of a catch-up vaccination campaign. While this is a small study, this is an important topic and vaccine data and efficacy from the Greater Mekong Subregion, particularly Myanmar, Laos, and Cambodia, is lacking and should be published.

Response: We thank the reviewers for their positive responses.

MAJOR COMMENT:

The authors point out the main limitation of the study, which is cross-sectional nature, and should be better emphasized in the abstract. The duration of nAb response will be critical to follow up.

Response: Thank you very much for your very kind and valuable comments We already added as the limitations of the study as it was just descriptive study and we did not investigate the persistence of antibody. We will continue further studies to understand the duration of neutralizing antibody response. Following your suggestion, we added the limitations at the revised abstract. We will follow up the students will investigate the persistence of neutralizing antibody among vaccinated students in the future studies. (Page-1, Line-24-25)

Were all the children vaccinated at the same time or could you model the antibody decay by using different vaccination dates of the children? (e.g. 50 children were vaccinated 6 months prior to blood draw and somewhere vaccinated 2 months prior to blood draw, etc).

Response: In this study, we recruited the students who received vaccines during the catch-up campaign. All students received the vaccines at the same time (within two days). As the first phase of catch-up vaccine campaign, JEV vaccination was done at school as mass campaign and all students except who fit in contraindications criteria was received the vaccine. Therefore, we could not analyses the data. Thank you very much for your expert and kind suggestion. We added more details about the sample population for better understating to the readers. (Page-2, Line-65-67)

There should be some type of sample size calculation so the reader can gauge the precision of your estimates given that 198 children is not a very large number, especially if this will then extrapolated nationally.

Response: We calculated the sample size using the formula, N =Z2P (1-P)/ d2 . We assumed the proportion of presence of neutralizing antibody among vaccinated students would be 40% and precision was taken as 0.07% and 95% Confidence Interval. We added the sample size calculation formula at the revised manuscript. (Page-2, Line-67-69)

The 95% CIs are large in the table and are an indicator that there is not very much precision so it would be nice to see these calculations quantified somewhere.

Response: We found our 95% CI calculation error, and corrected in Table 1.

The 95% CI are large at some group due to some outlier cases (eg. only one case of high neutralizing antibody titers 1:5,120). We did not remove the outlier cases during analysis and therefore the 95% CI are large.

Otherwise, I only have some minor comments for review:

Line 160-162: example - nomenclature for nAB - "students had greater than 10 NAbs against JEV" - it should be titers of 1:10 - it is incorrectly written in a few places like this.

Response: Following the suggestion, we corrected the mistakes. (Page-3,4,5, Line-119, 216, 268, 269, 271, 272)

Line 178: for those positive on ELISA but negative on nABs, what is the likelihood of non-neutralizing antibodies being responsible? It talks about IgG being cross-reactive but more specificity is needed - is it presumably NS1 IgG that is cross reactive with other flavivirus NS1? It says purified JEV antigen in the Methods - is it ELISA against whole virus?

Response: We used purified whole virus as coating antigen for detection of IgG Ab in this study. Therefore, antibodies may be cross reactive with other flaviviruses due to high similarities of antigens among them. For clear understanding of the readers, we added the word “whole virus” at the revised manuscript. (Page-2, Line-78)

Line 180: there is an extra "."

Response: We removed the extra “.”.(Page-7, Line-288)

Reviewer 2 Report

The authors studied the effectiveness of the SA-14- 11 14-2 live-attenuated JEV vaccine for the Japanese encephalitis virus (JEV). A total of 198 students who had received vaccines were recruited, and 15 single-time investigations of anti-JEV IgG and neutralizing antibodies against wild-type JEV were determined. Interestingly, all students tested showed positive results on the anti-JEV IgG ELISA, and 87% of the 18 students had neutralizing antibodies against JEV six months after immunization. The novelty of this study is that it is the first report of JEV vaccination among children in Myanmar.

The study is well-written, and the results are convincing, and the conclusions are well-raised. Only mirror things should be addressed.

In the statistical section, describe if the normality test was used. Describe the use of the U-Mann Whitney test.

There is a typo error in table 1, correct 7876 for 7.876.

Author Response

The authors studied the effectiveness of the SA-14- 11 14-2 live-attenuated JEV vaccine for the Japanese encephalitis virus (JEV). A total of 198 students who had received vaccines were recruited, and 15 single-time investigations of anti-JEV IgG and neutralizing antibodies against wild-type JEV were determined. Interestingly, all students tested showed positive results on the anti-JEV IgG ELISA, and 87% of the 18 students had neutralizing antibodies against JEV six months after immunization. The novelty of this study is that it is the first report of JEV vaccination among children in Myanmar.

Response: We thank the reviewer for giving the summary of our study.

The study is well-written, and the results are convincing, and the conclusions are well-raised. Only mirror things should be addressed.

Response: We thank the reviewers for their positive responses.

In the statistical section, describe if the normality test was used. Describe the use of the U-Mann Whitney test.

Response: In this study, we checked our data were normal distributed or not and we noted there was skew distribution. Therefore, we used Mann-Whitney U test. We added the reviewer suggestions at the revised manuscript. (Page-3, Line-110-111)

There is a typo error in table 1, correct 7876 for 7.876.

Response: We corrected the typo error. (Table-1)